Comparative transcriptome analysis to identify the important mRNA and lncRNA associated with salinity tolerance in alfalfa

Yang Gaimei 1
Li Zhengyan 2
Rong Mengru 1
Yu Rugang 1 yurugang@126.com
Zhang Qiting 1
Wang Guoliang 3 wangguoliang@126.com
Xu Zhiming 2
Du Xueling 1
Xu Xian 1
1 College of Life Sciences, Huaibei Normal University , Huaibei, Anhui , China
2 Animal Husbandry and Veterinary Research Institute, Anhui Academy of Agricultural Sciences , Hefei, Anhui , China
3 Institute of Leisure Agriculture, Shandong Academy of Agricultural Sciences , Jinan, Shandong , China
Naithani Sushma
Electronic publication date: 2024 Oct 16
Publication date: 2024
Volume: 12
Electronic Location ID: e18236
Received 2024 Jun 6; Accepted 2024 Sep 15
Copyright: © 2024 Yang et al.
Copyright year: 2024
Copyright holder: Yang et al.
License: This is an open access article distributed under the terms of the Creative Commons Attribution License, which permits unrestricted use, distribution, reproduction and adaptation in any medium and for any purpose provided that it is properly attributed. For attribution, the original author(s), title, publication source (PeerJ) and either DOI or URL of the article must be cited.
License URL: https://creativecommons.org/licenses/by/4.0/

Keywords: Medicago sativa, Comparative transcriptomics, Salinity stress, lncRNA, mRNA

Funding: Anhui Provincial Key Research and Development Program Plan 202004a06020046 Earmarked Fund for China Agriculture Research System CARS-34 Science Research Foundation of the Education Bureau of Anhui Province of China 2022AH050385 This work was supported by the Anhui Provincial Key Research and Development Program Plan (No. 202004a06020046), the Earmarked fund for China Agriculture Research System (CARS-34) and the Science Research Foundation of the Education Bureau of Anhui Province of China (2022AH050385). The funders had no role in study design, data collection and analysis, decision to publish, or preparation of the manuscript.

==============================
Salinity represents a fatal factor affecting the productivity of alfalfa. But the regulation of salinity tolerance via lncRNAs and mRNAs remains largely unclear within alfalfa. For evaluating salinity stress resistance-related lncRNAs and mRNAs within alfalfa, we analyzed root transcriptomics in two alfalfa varieties, GN5 (salinity-tolerant) and GN3 (salinity-sensitive), after treatments with NaCl at 0 and 150 mM. There were altogether 117,677 lncRNAs and 172,986 mRNAs detected, including 1,466 lncRNAs and 2,288 mRNAs with significant differential expression in GN5150/GN50, GN3150/GN30, GN50/GN30, and GN5150/GN3150. As revealed by GO as well as KEGG enrichment, some ionic and osmotic stress-associated genes, such as HPCA1-LRR, PP2C60, PP2C71, CRK1, APX3, HXK2, BAG6, and ARF1, had up-regulated levels in GN5 compared with in GN3. In addition, NaCl treatment markedly decreased CNGC1 expression in GN5. According to co-expressed network analyses, six lncRNAs (TCONS_00113549, TCONS_00399794, TCONS_00297228, TCONS_00004647, TCONS_00033214 and TCONS_00285177) modulated 66 genes including ARF1, BAG6, PP2C71, and CNGC1 in alfalfa roots, suggesting that these nine genes and six lncRNAs probably facilitated the different salinity resistance in GN5 vs. GN3. These results shed more lights on molecular mechanisms underlying genotype difference in salinity tolerance among alfalfas.

Introduction

Soil salinization accounts for an important factor restraining crop productivity worldwide (Zhao et al., 2021; Wang et al., 2022), including alfalfa (Medicago sativa L.). Salinity affects nearly 1/2 of irrigated and 1/5 of cultivated lands, besides, such values keep elevating (Gupta & Huang, 2014; Shabala, 2013). Therefore, it is the efficient and economical way to develop and plant high salinity-tolerant crop varieties for improving and utilizing salinized soils (Zörb, Geilfus & Dietz, 2019; Wang et al., 2022). In this regard, comprehensive elucidation of mechanisms underlying salinity resistance and identification of corresponding genes and genotypes are necessary.

Alfalfa, a key perennial forage crop showing high quality and productivity, is planted worldwide. Relative to additional forage plants, alfalfa shows moderate salinity resistance (Bertrand et al., 2015), showing a wide variation among varieties (Benabderrahim, Guiza & Haddad, 2020; Tavakoli et al., 2019; Yu et al., 2022). It is a critical goal to improve salinity resistance during alfalfa breeding. Previously, salinity resistance in alfalfa exhibits positive relation to increased shoot K+/Na+ ratio and Na+ exclusion (Tavakoli et al., 2019; Yu et al., 2021), enhanced antioxidant enzyme activity (Benabderrahim, Guiza & Haddad, 2020; Cen et al., 2020; Ashrafi et al., 2015), osmotic adjustments (Bertrand et al., 2020, 2016) and membrane protection (Ashrafi et al., 2015), demonstrating that the above factors probably have critical effects on analyzing different salinity resistance among varieties. Existing research has illustrated certain physiological mechanisms underlying salinity resistance of different alfalfa varieties, but its molecular mechanism remains largely unclear.

Plants evolve different strategies for survival upon salinity stress (Wang et al., 2022; Zhao et al., 2021). Consequently, salinity resistance represents the complicated trait regulated via different genes related to salt perception, signaling, osmotic adjustment, transcription modulation, reactive oxygen species (ROS) removal, and ion homeostasis. For instance, pathways like receptor-like protein kinase (RLK) (Huang & Joosten, 2024; Wang et al., 2021), calmodulin (CaM) (Ketehouli et al., 2022), Ca2+-dependent protein kinase (CDPK) (Ketehouli et al., 2022; Tao & Lu, 2013), calcineurin B-like protein (CBL)/CBL-interacting protein kinase (CIPK) (Fan et al., 2019; Miranda et al., 2017), mitogen-activated protein kinase (MAPK) (Wang et al., 2022; Zhu, 2016), abscisic acid (ABA) (Zhao et al., 2021) and ROS pathways (Kale & Irmale, 2022; Zelm, Zhang & Testerink, 2020) are associated with ionic, osmotic, and ROS homeostasis. Up-regulation of genes related to transcription factors (TFs) (i.e., basic helix–loop–helix 18 (MxbHLH18) and MxWRKY53) (Liang et al., 2022; Han et al., 2022), antioxidases (i.e., superoxide dismutase (AhCuZnSOD) and ascorbate peroxidase 3 (AtAPX3)) (Negi et al., 2017; Wang, Zhang & Allen, 1999), osmotic regulation (i.e., hexokinases 1 (MdHXK1) and proline-rich protein 6 (AtBAG6)) (Arif et al., 2021; Sun et al., 2018) and ion transporter (i.e., sodium/hydrogen exchanger 7 (OsNHX7) and cyclic nucleotide-gated ion channel 1 (CNGC1)) (Awaji et al., 2020; Zhao et al., 2022a) can alter salinity resistance of transgenic plants. Some genes related to response to salinity stress, including MsCBL4 (An et al., 2020), Protein Phosphatase 2C Alpha 1 (MsPP2CA1) (Dong et al., 2019) and peroxidase (MsPOD) (Teng et al., 2016), are identified in alfalfa.

RNA sequencing (RNA-seq) has been developed as the potent approach to discover potential genes and pathways related to salinity resistance, which is extensively adopted for numerous plants (Shams et al., 2023; Duan et al., 2022; Mu et al., 2021). Long non-coding RNAs (lncRNAs) are functional RNAs 200 nt long, which can not code proteins and they have helped understand eukaryote transcriptome (Zhao et al., 2022c). They have crucial functions in regulating post-transcriptional translation and transcription levels of genes, and are widely related to plant response to salinity (Jiang et al., 2022; Kaashyap et al., 2022; Qin et al., 2017), however, functions of lncRNAs are rarely explored in alfalfa.

The present work conducted comparative transcriptomic analysis for identifying differentially expressed (DE) lncRNAs and mRNAs within roots in Gannong No. 5 (GN5, salinity tolerance) and Gannong No. 3 (GN3, salinity sensitive) after NaCl and control treatments. As suggested by co-expression analysis, nine genes and six lncRNAs probably improving salinity resistance of alfalfa. Our findings shed more lights on molecular mechanism and genetic foundation for salinity stress responses of alfalfa.

Materials and Methods

Plant materials and salinity treatments

Two alfalfa varieties, GN5 (salinity tolerance) and GN3 (salinity sensitive) were used in this work according to our prior results (Yu et al., 2022). Two-weeks-old seedlings of every variety were subjected to 0 and 150 mM NaCl treatments separately. After 2 weeks of treatment, roots were obtained in 0 and 150 mM NaCl treatment seedlings for RNA-seq and RT-qPCR (two and three biological replicates, respectively) (Du et al., 2021).

RNA isolation, library establishment, and sequencing

By adopting TRIzol regent (Invitrogen, Waltham, MA, USA), we isolated total RNAs from roots. Eight transcriptome libraries, namely, GN50_1, GN50_2, GN5150_1, GN5150_2, GN30_1, GN30_2, GN3150_1 and GN3150_2 based on two duplicate RNA samples under 0 and 150 mM NaCl treatments were employed for high-throughput sequencing by NovaSeq 6000 platform (Novogene Bioinformatics Technology Co., Ltd., Beijing, China), and the 150 bp paired-end reads were obtained. Removing the adaptor, low-quality, and poly-N readers from raw reads, the clean reads were matched to alfalfa reference genome by Hisat2 (Kim, Langmead & Salzberg, 2015). Afterwards, StringTie software (Pertea et al., 2015) and cuffmerge (Trapnell et al., 2010) were utilized to assemble mapped reads for every sample into transcripts.

CPC (Kong et al., 2007), CNCI (Sun et al., 2013) and PFAM (Mistry et al., 2021) were adopted for predicting transcripts’ coding potentials. The sequences with no coding potential were defined as the novel lncRNAs. Meanwhile, mRNAs and lncRNAs expression abundances were determined with StringTie software and converted into FPKM data.

DE mRNAs and lncRNAs discovery and functions

DE analysis was carried out in four groups (GN50, GN5150, GN30 and GN3150) with edgeR R package, while DE lncRNAs and mRNAs were selected upon the threshold of adjusted P-value < 0.05. Later, DE mRNAs and lncRNAs were performed to GO as well as KEGG enrichment with GOseq 3.3.2 (https://www.bioinfo-scrounger.com/archives/227/) and KOBAS v2.0 (https://www.jianshu.com/p/90d67997b51b), respectively. We deemed GO terms together with KEGG pathways (Kanehisa & Goto, 2000; Kanehisa et al., 2023) satisfying P < 0.05 as significant enrichment. R (pheatmap) was applied in obtaining the clustered heatmap.

Target genes of lncRNAs and their functions

Functions of DE lncRNAs were estimated in line with co-expression relation of lncRNAs with mRNAs. The lncRNA–mRNA co-expressed pairs were selected upon thresholds of Pearson’s correlation coefficient >0.95 or <−0.95 and P-value < 0.05. Cytoscape software was utilized to establish a co-expression network involving six important DE lncRNAs and the target-related DE mRNAs.

Real-time quantitative polymerase chain reaction (RT-qPCR) validation

Total RNA from root samples were used to synthesis cDNA using Prime Script®RT reagent Kit (Takara, Dalian, China). The ABI 7300 (Applied Biosystems, Foster City, CA, USA) was used for RT-qPCR with the SYBR Premix EX Taq kit (Takara) (Du et al., 2021). All primers utilized for RT-qPCR can be obtained from Table S1.

Results

Summary of raw sequence data

RNA-seq data were examined in eight root samples (2 genotypes × 2 treatments × 2 biological duplicates), resulting in 80,691,030–106,774,256 raw data as well as 80,182,796–106,085,034 clean data (Table S2). Hisat2 software was later used to map clean reads in every sample to the genome of autotetraploid-cultivated alfalfa, with the mean genome mapping rate of 70.96% (Table S2).

LncRNAs and mRNAs identification in alfalfa

There were altogether 117,677 candidate novel lncRNAs (Fig. 1A) obtained in eight libraries, which included 97,669 intergenic lncRNAs (lincRNAs) (83.0%), 16,845 antisense lncRNAs (14.3%), as well as 3,163 sense overlapping lncRNAs (2.7%) (Fig. 1B). In addition, we also obtained 172,986 genes, which included 164,632 known mRNAs and 8,354 new mRNAs. According to our results, new lncRNAs had small size, with less exons and open reading frames compared with mRNAs (Figs. 1C–1E).

Figure 1 Characterization of lncRNAs and mRNAs in alfalfa root tissue.

Coding potential analysis via CPC, PFAM and CNCI (A). The classification of identified novel lncRNAs (B). Density distribution diagram showing the expression features of exon number (C), length (D) and opening reading frame (ORF) (E) of novel lncRNAs and mRNAs in alfalfa root tissue.

DE analysis on lncRNAs and mRNAs

We conducted pairwise comparison of lncRNAs and mRNAs in GN5 vs. GN3 (GN50/GN30 and GN5150/GN3150) or in control vs. NaCl-treated samples for every variety (GN5150/GN50 and GN3150/GN30). Adjusted P-value < 0.05 was applied in determining DE lncRNAs and mRNAs. There were altogether 2,288 DE mRNAs (Table S3) and 1,466 DE lncRNAs (Table S4) obtained by four comparisons. There were 784 DE mRNAs and 504 DE lncRNAs obtained in two varieties under control condition, whereas 895 DE mRNAs and 613 DE lncRNAs after NaCl treatment (Figs. 2A, 2B). Between GN50/GN30 and GN5150/GN3150, we obtained 87 common mRNAs (Fig. 2C) and 41 common lncRNAs (Fig. 2D). There were 849 DE mRNAs (447 up- whereas 402 down-regulated) and 488 DE lncRNAs (243 up- whereas 245 down-regulated) obtained in GN5 after NaCl treatment. In comparison, 658 DE mRNAs (327 up- whereas 331 down-regulated) (Fig. 2A) and 316 DE lncRNAs (139 up- whereas 177 down-regulated) (Fig. 2B) were identified in GN3. Between GN5150/GN50 and GN3150/GN30, there were just 71 shared DE mRNAs (Fig. 2C) as well as 22 shared DE lncRNAs (Fig. 2D) after NaCl treatment of two varieties. Moreover, for evaluating DE mRNAs and DE lncRNAs expression, we carried out hierarchical clustering with FPKM values for mRNAs and lncRNAs after 0 and 150 mM NaCl treatment in GN5 and GN3. As a result, genotype probably significantly affected mRNA and lncRNAs levels, because GN5150 and GN50 generated one cluster, whereas GN3150 and GN30 produced another one (Figs. 3A, 3B). Obvious DE could be seen in two varieties after salinity stress (Figs. 3A, 3B), suggesting the possible effects of DE mRNAs and DE lncRNAs on salinity stress responses.

Figure 2 The number of DE mRNAs (A) and lncRNAs (B) between pairwise of GN50, GN5150, GN30 and GN3150 groups. Venn diagrams analysis of mRNAs (C) and (D) lncRNAs in any two groups.

Figure 3 Hierarchical clustering of DE mRNAs (A) and lncRNAs (B) under control and NaCl treatment in two alfalfa varieties.

Functions of DE mRNAs

We performed GO for analyzing major functions of DE mRNAs. There were altogether 100, 149, 93 and 150 GO terms (P-value < 0.05) acquired in GN5150/GN50, GN3150/GN30, GN50/GN30 and GN5150/GN3150, separately, and these terms can be classified as three categories (biological process, cellular component and molecular function) (Table S5). With regard to NaCl-responsive DE mRNAs, those three most significantly related GO categories in GN5 included ‘oxidation-reduction process’, ‘carbohydrate metabolic process’, and ‘cofactor binding’ (Fig. 4A), while those in GN3 included ‘metabolic process’, ‘primary metabolic process’ and ‘single-organism metabolic process’ (Fig. 4B); moreover, there were four common GO categories in two varieties, including ‘3′-5′ DNA helicase activity’, ‘ATP-dependent 3′-5′ DNA helicase activity’, ‘hydrolase activity, acting on glycosyl bonds’, and ‘NAD+ nucleosidase activity’ (Table S5). In terms of DE mRNAs after NaCl treatment, those three most significantly enriched GO terms in GN5 and GN3 included ‘cellular process’, ‘single-organism cellular process’ and ‘nitrogen compound metabolic process’ (Fig. 4D).

Figure 4 Gene ontology (GO) functional enrichment of DE mRNAs in alfalfa.

The top 15 abundant categories of GO terms for GN5150/GN50 (A), GN3150/GN30 (B), GN50/GN30 (C) and GN5150/GN3150 (D).

Upon the threshold of P < 0.05, we obtained altogether 7, 5, 4 and 5 obviously involved KEGG pathways from GN5150/GN50, GN3150/GN30, GN50/GN30 and GN5150/GN3150, separately (Table 1). Two most significant pathways enriched by NaCl-responsive DE mRNAs included ‘sesquiterpenoid and triterpenoid biosynthesis’ (ko00909) and ‘carotenoid biosynthesis’ (ko00906) in GN5, whereas ‘Amino sugar and nucleotide sugar metabolism’ (ko00520) and ‘Aminoacyl-tRNA biosynthesis’ (ko00970) in GN3. There was no pathway markedly involved in two varieties. Pathways enriched by DE mRNAs after NaCl treatment in GN5 and GN3 included ‘tryptophan metabolism’ (ko00380), ‘nicotinate and nicotinamide metabolism’ (ko00760), ‘biotin metabolism’ (ko00780), ‘2-oxocarboxylic acid metabolism’ (ko01210), and ‘nucleotide excision repair’ (ko03420) (Table 1).

Table 1 Significantly enriched KEGG pathways for the DE mRNAs.

Pathway term	DE mRNAs tested	P-value	Pathway ID	
GN5150/GN50	GN3150/GN30	GN50/GN30	GN5150/GN3150	GN5150/GN50	GN3150/GN30	GN50/GN30	GN5150/GN3150	
Sesquiterpenoid and triterpenoid biosynthesis	8	2	2	#N/A	0.0000	0.0713	0.0962	#N/A	ko00909	
Carotenoid biosynthesis	7	#N/A	1	2	0.0033	#N/A	0.7860	0.5314	ko00906	
Cysteine and methionine metabolism	10	1	5	3	0.0047	0.9248	0.2131	0.6908	ko00270	
Peroxisome	9	3	4	6	0.0165	0.5145	0.4157	0.1823	ko04146	
Fatty acid degradation	6	2	2	4	0.0258	0.4670	0.5606	0.1786	ko00071	
Inositol phosphate metabolism	6	4	5	2	0.0311	0.0908	0.0555	0.6566	ko00562	
N-Glycan biosynthesis	5	3	2	4	0.0432	0.1578	0.4776	0.1218	ko00510	
Amino sugar and nucleotide sugar metabolism	9	14	12	3	0.1399	0.0002	0.0077	0.9284	ko00520	
Aminoacyl-tRNA biosynthesis	1	9	5	3	0.9902	0.0106	0.4032	0.8483	ko00970	
Glycosaminoglycan degradation	1	3	1	1	0.4308	0.0111	0.3924	0.4332	ko00531	
Stilbenoid, diarylheptanoid and gingerol biosynthesis	1	3	1	1	0.5363	0.0236	0.4931	0.5390	ko00945	
Purine metabolism	14	12	8	13	0.0592	0.0303	0.4878	0.1060	ko00230	
Propanoate metabolism	1	1	4	1	0.6762	0.5663	0.0229	0.6789	ko00640	
Galactose metabolism	2	3	6	1	0.7403	0.3157	0.0365	0.9284	ko00052	
Proteasome	6	3	6	5	0.0636	0.3260	0.0393	0.1493	ko03050	
Tryptophan metabolism	1	1	1	5	0.6762	0.5663	0.6310	0.0086	ko00380	
Nucleotide excision repair	5	6	5	11	0.6481	0.2322	0.5405	0.0314	ko03420	
Biotin metabolism	#N/A	#N/A	#N/A	3	#N/A	#N/A	#N/A	0.0366	ko00780	
2-Oxocarboxylic acid metabolism	3	1	3	6	0.4320	0.8260	0.3577	0.0416	ko01210	
Nicotinate and nicotinamide metabolism	1	#N/A	1	3	0.5119	#N/A	0.4696	0.0434	ko00760	

Osmotic and ionic stress-associated DE mRNAs

In this study, a total of 36, 10, 21, 63 and 24 DE mRNAs showed high similarity with signaling pathways (Table S6), osmolyte synthesis (Table S7), ROS-scavenging (Table S8), transcription factors (Table S9), and transporters (Table S10) related genes, respectively. They were up or down-regulated in four comparisons. Typically, eight osmotic and ionic stress-associated genes, namely, HPCA1-LRR, PP2C60, PP2C71, CRK1, APX3, HXK2, BAG6, and ARF1 exhibited up-regulation in GN5 after NaCl treatment, with increased levels in GN5 relative to GN3 (Table 2). CNGC1 showed specific expression in GN5, which markedly decreased after NaCl treatment (Table 2). After NaCl treatment, these nine genes did not show any expression or change in expression in GN3 with the exception of CRK1 and HXK2 (down-regulation) (Table 2). Based on prior studies, these nine genes were associated with salinity resistance of GN5, which should be further investigated to examine effects of lncRNAs on salinity stress responses of alfalfa.

Table 2 Critical DE mRNAs involved in the salt tolerance of alfalfa from this study and their putative roles from the literature.

Gene ID	Gene description	Gene name	log2 FoldChange	Effect of salt tolerance	Reference	
GN3150/GN30	GN5150/GN50	GN50/GN30	GN5150/GN3150	
Signal intermediates									
MS.gene80177	Probable protein phosphatase 2C 60	PP2C60	#N/A	13.757349*	#N/A	13.789450*	Positive	Zhang et al. (2017)	
MS.gene002188	Probable protein phosphatase 2C 71	PP2C71	−5.742524	11.476081*	−5.675029	11.522385*	Positive	Zhang et al. (2017)	
MS.gene017713	CDPK-related kinase 1	CRK1	−13.65268*	14.601374*	−13.608622*	14.648635*	Positive	Tao & Lu (2013)	
MS.gene047377	leucine-rich repeat receptor-like protein kinase	HPCA1	#N/A	3.358444	6.650548	10.046898*	Positive	Wu et al. (2020)	
Antioxidant									
MS.gene59642	L-ascorbate peroxidase 3, peroxisomal	APX3	−7.616087	9.735014*	−7.547241	9.780866*	Positive	Wang, Zhang & Allen (1999)	
Osmotic adjustment substance									
MS.gene64012	Hexokinase-2	HXK2	−9.588875*	9.596857*	−9.530094*	9.644086*	Positive	Sun et al. (2018)	
MS.gene27108	Large proline-rich protein BAG6	BAG6	−1.839044	6.459654*	−1.932951	6.359187*	Positive	Arif et al. (2021)	
Transcription factors									
MS.gene002000	Auxin response factor 1	ARF1	#N/A	12.936201*	#N/A	12.999012*	Positive	Wang et al. (2020)	
Ion transporters									
MS.gene06955	Cyclic nucleotide-gated ion channel 1	CNGC1	#N/A	−13.7206383*	13.70705296*	#N/A	Negative	Zhao et al. (2022a)	
Note:

* Indicates the genes significant differential expression between two groups.

mRNA-lncRNA co-expression network

For identifying critical lncRNAs associated with salinity tolerance in alfalfa roots, we obtained the lncRNAs target genes based on the lncRNAs-mRNAs expression correlation, and then we searched for the target-associated DE lncRNAs from the above nine salinity-tolerance related genes. As a result, we obtained four genes (ARF1, BAG6, PP2C71, and CNGC1) together with 6 target-associated DE lncRNAs in the mRNA–lncRNA co-expression network (Table 3). According to DE analysis (Table S11), we established the lncRNA–mRNA network including 6 DE lncRNA nodes, 66 DE mRNA nodes (green and yellow dots separately in Fig. 5) and 109 edges. Diverse edges stand for diverse interrelationships. For instance, TCONS_00004647 shows co-expression with 24 among those 67 obtained DE mRNAs and with a majority of the obtained DE lncRNAs (Fig. 5). Moreover, TCONS_00297228 exhibited co-expression with 23 mRNAs (Fig. 5).

Table 3 The co-expression relationships of critical DE mRNAs and their target-associated DE lncRNA in this study.

lncRNA_ID	mRNA Gene ID	mRNA Gene name	Pearson_correlation	P-value	log2 FoldChange	
GN3150/GN30	GN5150/G50	GN50/GN30	GN5150/GN3150	
TCONS_00113549	MS.gene002000	ARF1	0.99797333	2.07792E−08	−3.147748936	11.61438681*	−3.085178	11.718432*	
TCONS_00399794	MS.gene002000	ARF1	0.971862984	5.4521E−05	−7.426071578	11.04856595*	−7.339668	11.146698*	
TCONS_00297228	MS.gene27108	BAG6	0.969567658	6.89E−05	#N/A	2.216445497	6.533604	8.849388*	
TCONS_00004647	MS.gene27108	BAG6	0.969336531	7.04E−05	#N/A	4.478766193	5.163015	9.737429*	
TCONS_00297228	MS.gene002188	PP2C71	0.961009863	0.000143885	#N/A	2.216445497	6.533604	8.849388*	
TCONS_00004647	MS.gene002188	PP2C71	0.995166001	2.81374E−07	#N/A	4.478766193	5.163015	9.737429*	
TCONS_00033214	MS.gene06955	CNGC1	1	0	#N/A	−18.81176035*	18.796988*	#N/A	
TCONS_00285177	MS.gene06955	CNGC1	0.992123457086	0.00000121444497612	#N/A	−4.413748039*	11.425630*	7.084969	
Note:

* Indicates the genes significant differential expression between two groups.

Figure 5 The critical DE lncRNA and mRNA co-expression network in alfalfa.

The co-expression network consists of the six DE lncRNAs (green dots), correlated 66 mRNAs (yellow dots) and 109 edges.

RT-qPCR verification

For verifying RNA-seq data, we analyzed expression of nine DE mRNAs (PP2C71, CRK1, CRK26, PUB17, APX3, HXK2, ARF1, NAC90, and CNGC1) and three co-expressed DE lncRNAs (TCONS_00297228, TCONS_00033214, and TCONS_00285177) by RT-qPCR. As a result, many genes exhibited similar expression patterns to RNA-seq-based counterparts (Fig. 6). Typically, two genes APX3 and HXK2 exhibited inconsistent expression patterns in RT-qPCR compared with RNA-seq data (Fig. 6). Additionally, RT-qPCR detected the diverse expression levels of two genes (ARF1 and CNGC1) and three lncRNAs (TCONS_00297228, TCONS_00033214, and TCONS_00285177) in GN3, but they were not detected in RNA-seq data (Fig. 6). Such difference is probably related to the heterogeneous sensitivities of these two methods.

Figure 6 RT-qPCR analyses of nine genes and three lncRNAs under 0 and 150 mM NaCl treatments in roots of two alfalfa varieties.

Each bar represents the mean ± SE of triplicate assays. Values with different letters indicate significant differences at p < 0.05 according to Duncan’s multiple range tests.

Discussion

Possible DE mRNAs determining the different salinity resistance in GN5 and GN3 signaling-associated DE mRNAs

Signaling contributes to perceiving and transducing stress signals and activating defense mechanisms to facilitate plant survival upon salt stress (Wang et al., 2022). We obtained 36 DE mRNAs related to RLK, CDPK, CBL-CIPK, CaM, MAPK, and ABA pathways (Table S6). RLKs represented the upstream signaling molecules with high conservative degrees, which can modulate numerous defense processes (Huang & Joosten, 2024). Cysteine- and leucine-rich receptor-like kinases (CRKs and LRRs separately) account for two main RLK classes that are important for plant responses to abiotic stresses and cell death (Wang et al., 2021; Lin et al., 2020). In this work, we discovered 10 DE mRNAs enriched into RLK pathway (Table S6). Typically, CRK26 (MS.gene24334) gene showed down-regulation in GN5 after NaCl treatment, and its expression was not changed in GN3. Besides, its expression decreased in GN5 compared with GN3 after NaCl treatment (Table S6). CRK family members have diverse activities upon salt stress. Overexpression of PaCRK1 in Arabidopsis and sweet cherry enhanced tolerance to salt stress in transgenic plants (Zhao et al., 2022b), while Arabidopsis plants with AtCRK45 overexpression show lower salinity stress resistance (Zhang et al., 2013). More investigations are needed to examine whether CRK26 plays an important role in modulating GN5 salinity resistance. Additionally, there was one HPCA1 gene showing specific expression in GN5, which was up-regulated in GN5 compared with GN3 after NaCl treatment (Table 2). Moreover, HPCA1 gene is responsible for encoding one LRR kinase for mediating H2O2-mediated Ca2+ channel activation in stressed guard cells (Wu et al., 2020), suggesting the probable contribution of HPCA1 to different salinity resistance of two alfalfa varieties.

Calcium signals are the critical intercellular secondary messengers during diverse biological processes (Ketehouli et al., 2022). Calcium cascades including CaMs, CDPKs, CBLs, CIPKs and MAPKs have essential effects on plant adaptability to salinity (Ketehouli et al., 2022; Zhao et al., 2021; Zhu, 2016). We obtained altogether 16 DE mRNAs that encoded CML10, CIP111, PICBP, CAMTAs, CDPKs, CBL3, CIPK25, and MAPKs from four comparisons (Table S6), demonstrating that salinity stress might activate signal perception and transduction in alfalfa. Noteworthily, AtCRK1 (CDPK related kinase 1) has a positive effect on regulating salt/heat stress resistance of plants (Tao & Lu, 2013). We found that CRK1 expression decreased in GN3 after NaCl treatment, but its expression increased in GN5. Additionally, CRK1 expression increased in GN5 relative to GN3 after NaCl treatment (Table 2), indicating the potential effect of CRK1 gene on the different salinity resistance of two alfalfa varieties.

ABA has a critical role in biotic/abiotic stress responses in plants (Wang, Yu & Xie, 2020; Zhao et al., 2020). The classic ABA-PYL-PP2C-SnRK2 pathway has been extensively explored. The ABA receptor PYL/PYR/RCAR contributes to sensing ABA while suppressing PP2Cs, which exerts the positive regulation on its downstream sucrose nonfermenting 1-related protein kinase 2 (SnRK2) (Santiago et al., 2012). SnRK2 is responsible for phosphorylating TF ABFs for regulating stress-responsive gene levels (Zhao et al., 2021). According to our results, 1 ABF2, 2 SnRK1 (KING1 and KINB2) and 9 PP2Cs had changed levels from four comparisons (Table S6). Typically, 3 PP2Cs (PP2C60: MS.gene054537, PP2C60: MS.gene80177, PP2C71) and 4 PP2Cs (PP2C6, 2 PP2C55, PP2C60: MS.gene054537) exhibited up-regulation in GN5 and GN3 separately after NaCl treatment. After NaCl treatment, PP2C33, PP2C71, PP2C60: MS.gene054537, ABF2, and KINB2 were up-regulated in GN5 relative to GN3. In GN5, NaCl treatment did not affect PP2C33, ABF2 and KINB2 expression. Many clade A PP2C members negatively regulated ABA pathway, but according to Zhang et al. (2017), BdPP2CA6 exerted positive modulation on ABA and stress pathway within seedlings of transgenic Arabidopsis plants, suggesting the positive regulation of PP2C71 and PP2C60 on ABA and stress pathway, thereby enhancing salinity resistance of GN5.

Osmoregulation-associated DE mRNAs

Osmotic stress may reduce water availability to plants, and it may be caused by salinity. Plants evolve defense mechanisms for achieving osmotic adjustment, including accumulating compatible solutes/osmolytes like glycine betaine, polyamines, proline, trehalose or soluble sugars into chloroplast and cytosol (Zelm, Zhang & Testerink, 2020). We obtained 10 osmolyte synthesis-associated DE mRNAs from four comparisons (Table S7). NaCl treatment enhanced four gene levels, namely, BAG6, PERK11, HXK2, and SPEA, and reduced PERK9 gene expression in GN5. After NaCl treatment, one gene (HXK1) in GN3 had increased expression, while 2 (HXK2 and TPS5) had decreased expression. Based on the above findings, osmolyte synthase-associated mRNAs were probably related to NaCl response within roots in GN5 and GN3. In GN5, NaCl treatment increased two genes (BAG6 and HXK2) expression relative to that in GN3 (Table 2). AtBAG6 (Arif et al., 2021) and MdHXK1 (Sun et al., 2018) overexpression enhanced salinity resistance of transgenic plants. But the effect of HXK2 on plant salinity resistance is still unclear. Based on the above findings, BAG6 may result in salinity resistance difference among different alfalfa varieties.

Upon osmotic stress, ROS can lead to oxidative stress and cell death as the toxic products (Zelm, Zhang & Testerink, 2020). ROS are scavenged via different enzymatic/nonenzymatic antioxidants (Kale & Irmale, 2022). As revealed by transcriptomic analysis, we obtained 21 DE mRNAs associated with antioxidant defense system (like APXs, PODs, SODs, MDARs, GSTs, TXNL4B and GRXC11) from four comparisons (Table S8). Typically, eight of them showed up-regulation in GN5 after NaCl treatment, while just two exhibited up-regulation in GN3 (Table S8), demonstrating the differences in ROS scavenger levels of these two alfalfa varieties, probably resulting in different salinity resistance. In GN5, NaCl treatment increased APX3 expression (MS.gene59642) compared with that in GN3, while its expression was not changed in GN3 following NaCl treatment (Table 2). AtAPX3 overexpression promotes oxidative stress resistance of transgenic tobacco (Wang, Zhang & Allen, 1999), indicating the important effect of APX3 gene on the different salinity resistance of two alfalfa varieties.

TF-associated DE mRNAs

TFs are crucial for biotic/abiotic stress responses in plants through modulating some downstream stress-responsive genes (Zelm, Zhang & Testerink, 2020; Zhao et al., 2020). The present work obtained 63 DE mRNAs associated with 10 TF families (Table S9), of them, ARF1 exhibited up-regulation in GN5150/GN50 and GN5150/GN3150, and it was unchanged in GN3150/GN30 (Table 2). Plants with PdPapARF1 overexpression exhibited the positive regulatory factor for enhancing poplar development and defense responses, similar to the effect of inoculation with Trichoderma asperellum (Wang et al., 2020), indicating ARF1 gene as the important factor for analyzing different salinity resistance of two alfalfa varieties.

Ionic transporters-associated DE mRNAs

Certain transporters like NHXs, ALMTs, CNGCs and KEAs are validated to be associated with sequestration or exclusion of cellular Na+, Cl−, and K+ within plants and can increase salinity resistance (Wang et al., 2019; Zhao et al., 2021). We discovered that 24 DE mRNAs were related to Na+, Cl−, and K+ transport (Table S10), which included 12 NaCl-responsive DE mRNAs in GN5, with five showing up-regulation whereas seven showing down-regulation. In comparison, in GN3, just there were just seven Na+, Cl−, and K+ transporters deemed as DE mRNAs. Additionally, the expression of most transporter-encoding DE mRNAs (71.43%) increased in GN5 relative to GN3 after control treatment (Table S10). Consequently, it was assumed that the salinity resistance mechanism in GN5 was the potential inherent trait. CNGC1 exhibited down-regulation in GN5 after NaCl treatment, which was up-regulated in GN5 compared with GN3 after control treatment (Table 2). In Arabidopsis, AtCNGC10 has a negative effect on regulating salinity resistance through regulating Na+ transport (Jin et al., 2015). GhCNGC1&18-silenced in cotton enhanced salt tolerance in transgenic plants (Zhao et al., 2022a), suggesting the possible involvement of CNGC1 in different salinity resistance of two alfalfa varieties.

LncRNAs probably markedly associated with salt resistance of alfalfa via lncRNA-mRNA co-expression analysis

LncRNAs are related to modulating different plant biological activities (Zhao et al., 2022c; Sun et al., 2023). There are many salinity stress-associated lncRNAs detected from multiple plants like chickpea (Cicer arietinum) (Kaashyap et al., 2022), cassava (Manihot esculenta Crantz) (Xiao et al., 2019) and M. truncatula (Wang et al., 2015). Additionally, for some lncRNAs, including DRIR (Qin et al., 2017), and Ptlinc-NAC72 (Jiang et al., 2022), their molecular functions are related to salinity resistance. Nonetheless, for the time being, alfalfa lncRNA number, features and expression profiles under salinity stress are still unknown. In this study, we obtained altogether 117,677 new lncRNAs, including 488, 316, 504 and 613 exhibiting DE levels in GN5150/GN50, GN3150/GN30, GN50/GN30 and GN5150/GN3150 comparison, respectively (Table S4). Besides, we predicted possible functions in key lncRNAs according to our constructed mRNA–lncRNA co-expression network (Table S11 and Fig. 5). By analyzing nine important mRNAs and the co-expressed lncRNAs, we selected six lncRNAs probably related to salinity resistance in alfalfa (Table 3). Typically, TCONS_00113549 and TCONS_00399794 showed co-expression with 16 mRNAs, with a majority of them being identical (93.75%), including ARF1, ABCG31 and PCR12. They exhibited up-regulation in GN5150/GN50 and GN5150/GN3150. Based on the above findings, TCONS_00113549 and TCONS_00399794 probably had similar effects on salinity stress resistance of alfalfa. TCONS_00297228 exhibited co-expression with 23 mRNAs (i.e., CYP87A3, BAG6 and NAA50) and up-regulation in GN5150/GN3150. TCONS_00004647 displayed co-expression with 24 mRNAs (i.e., BAG6, NAA50 and PP2C71), and up-regulation in GN5150/GN3150. TCONS_00033214 showed co-expression with 13 mRNAs (i.e., CNGC1, CAX2 and MPK4), upregulation in GN50/GN30, and downregulation in GN5150/GN50. TCONS_00285177 showed co-expression with 17 mRNAs (i.e., CNGC1, CAX2 and CML10) and up-regulation in GN50/GN30 (Table S11). But functions of such lncRNAs are rarely reported and required further investigations.

Conclusion

In this study, we obtained a total of 488, 316, 504 and 613 DE lncRNAs and 849, 658, 784 and 895 DE mRNA in GN5150/GN50, GN3150/GN30, GN50/GN30 and GN5150/GN3150 comparisons, respectively. Among them, several DE mRNAs might participate in salinity tolerance of GN5 by regulating signaling, ROS removal, ion homeostasis and osmoregulation. Through the construction of the mRNA-lncRNA co-expression networks, crucial lncRNAs and mRNAs probably associated with difference in salinity tolerance between the two alfalfa varieties were selected. Collectively, the schematic model was put forward for depicting the salinity tolerance regulation network in alfalfa (Fig. 7). Our results shed novel lights on the salinity tolerance molecular regulation network in alfalfa.

Figure 7 The putative model of salinity tolerance in GN5 compared with GN3.

The red font represents the up-regulated in GN5150/GN50 or GN5150/GN3150; the green font represents the down-regulated in GN5150/GN50.

Supplemental Information

Supplemental Information 1 Calculation of the power analysis of lncRNA-seq and mRNA-seq.

Supplemental Information 2 Supplementary Tables.

Additional Information and Declarations

Competing Interests

Author Contributions

Data Availability

The authors declare that they have no competing interests.

Gaimei Yang performed the experiments, analyzed the data, prepared figures and/or tables, authored or reviewed drafts of the article, and approved the final draft.

Zhengyan Li performed the experiments, analyzed the data, prepared figures and/or tables, authored or reviewed drafts of the article, and approved the final draft.

Mengru Rong performed the experiments, authored or reviewed drafts of the article, and approved the final draft.

Rugang Yu conceived and designed the experiments, performed the experiments, analyzed the data, prepared figures and/or tables, authored or reviewed drafts of the article, and approved the final draft.

Qiting Zhang performed the experiments, authored or reviewed drafts of the article, revised manuscript, and approved the final draft.

Guoliang Wang conceived and designed the experiments, analyzed the data, authored or reviewed drafts of the article, revised manuscript, and approved the final draft.

Zhiming Xu performed the experiments, prepared figures and/or tables, revised manuscript, and approved the final draft.

Xueling Du performed the experiments, prepared figures and/or tables, revised manuscript, and approved the final draft.

Xian Xu performed the experiments, authored or reviewed drafts of the article, revised manuscript, and approved the final draft.

The following information was supplied regarding data availability:

The raw data is available at the Genome Sequence Archive at Beijing Institute of Genomics (China National Center for Bioinformation): CRA008016.

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
