# Peer review of "Comparative transcriptome analysis to identify the important mRNA and lncRNA associated with salinity tolerance in alfalfa"

_PeerJ, doi:10.7717/peerj.18236_

## Round 0.1 · original submission · Minor Revisions

Thanks for submitting your manuscript to PeerJ. Please address the comments of all three reviewers and revise the manuscript.

·

Basic reporting

1. Clear and unambiguous English used throughout the manuscript.
2. Literature references are adequate which provided sufficient information about the manyscript context but in discussion part of the results there is a lag of recent references to claim the hypothesis if possible Authors can add some more recent information in the discussion part.
3. Figures, Tables and Raw data are adequate with the given manuscript.

Experimental design

Research design and methodology fit adequately according to the hypothesis testing and all the ethical standards measures have been taken care of. Material and Methodologies have been described with sufficient information.

Validity of the findings

Authors claims of DE mRNA s determining the difference salinity stress level has been well established the expression of gene. All underlying data have been provided and conclusion are well stated.

Additional comments

Overall manuscript is well written and the results are well established with the testing hypothesis. Discussion part is well accompany with the output results. I think manuscript is well suited for the publication with the minor addition of some recent references.

Reviewer 2 ·

Basic reporting

No comment.

Experimental design

No comment.

Validity of the findings

No comment.

Additional comments

This article analyzed the RNA-seq data of salinity-tolerant and salinity-sensitive alfalfa to identify some mRNA and lncRNA associated with salinity tolerance. I think the results are interesting and the study is well organized. However, there are some issues need to be revised as below.
1. Please clarify the specific variety of GN5 and GN3.
2. The article “Medina CA, Samac DA, Yu LX. Pan-transcriptome identifying master genes and regulation network in response to drought and salt stresses in Alfalfa (Medicago sativa L.). Sci Rep. 2021 Aug 26;11(1):17203.” also identified some mRNA and lncRNA in response to high salinity in alfalfa. Compare to this article, what is the novelty of your study?

·

Basic reporting

Alfalfa, as a kind of perennial forage crop showing high quality and productivity, is distributed wildly in the world and shows moderate salinity resistance. Long non-coding RNAs (lncRNAs) have crucial functions in regulating post-transcriptional translation and transcription levels of genes. This study conducted comparative transcriptomic analysis for identifying differentially expressed lncRNAs and mRNAs within roots in the two varieties (salinity-tolerant and salinity-sensitive) after NaCl treatments. The aim is to explore the molecular mechanism for salinity stress responses of alfalfa.
There is a problem need to be revised as follow:
1 Line 87-88, ‘The present work conducted comparative transcriptomic analysis for identifying differentially expressed (DE) lncRNAs and mRNAs within roots’, but on information about mRNAs described in the Introduction.

Experimental design

No comment.

Validity of the findings

No comments.

Additional comments

There are some problems need to be revised as follow:
1 Line 89, the description of GN5 (salinity-tolerant) and GN3 (GN3, salinity-sensitive) is not easy to understand.
2 Line 95, ‘Two alfalfa varieties, GN5 (salinity tolerance) and GN3 (salinity sensitive)’ used in this study should be given the real variety name other than the code.
3 Line 134, ‘Summary of raw sequence date’, date should be revised to data.
4 Figure 1, the writing of lincRNA is wrong in the legend, and revised to lncRNA.
5 Line 361 or 586, ‘cultivars’ or Line 95 ‘varieties’ should be selected an used in the manu.
67 Some tables number such as table 1 in line 647 and table 2 in line 648 should be changed.

---

## Round 0.2 · accepted · Accept

Congratulations! Your manuscript is accepted!

·

Basic reporting

Manuscript is refined in a good manner after revision and fit for publication from my end.

Experimental design

The experimental designs are well written with proper cited refrence.

Validity of the findings

Study conclusion meet up the requirement of original research problem.

Reviewer 2 ·

Basic reporting

no comment

Experimental design

no comment

Validity of the findings

no comment

Additional comments

no comment

·

Basic reporting

No comment.

Experimental design

No comment.

Validity of the findings

No comment.

Additional comments

The author has revised the problems according to the review's comments.